# Shipping Temperature, Time and Media Effects on Equine Wharton’s Jelly and Adipose Tissue Derived Mesenchymal Stromal Cells Characteristics

**DOI:** 10.3390/ani12151967

**Published:** 2022-08-03

**Authors:** Eleonora Iacono, Aliai Lanci, Penelope Gugole, Barbara Merlo

**Affiliations:** 1Department of Veterinary Medical Sciences, University of Bologna, Via Tolara di Sopra 50, Ozzano Emilia, 40064 Bologna, Italy; aliai.lanci2@unibo.it (A.L.); penelopemaria.gugol2@unibo.it (P.G.); barbara.merlo@unibo.it (B.M.); 2Interdepartmental Centre for Industrial Research in Health Sciences and Technologies, University of Bologna, Via Tolara di Sopra 41/E, Ozzano Emilia, 40064 Bologna, Italy

**Keywords:** mesenchymal stromal cells, equine, adipose tissue, Wharton’s jelly, storage

## Abstract

**Simple Summary:**

Today, the use of horse adipose tissue and Wharton’s jelly-derived mesenchymal stromal cells in veterinary regenerative medicine represents a promising tool. Cells need to be isolated and expanded in vitro in the laboratory to obtain a sufficient amount for clinical application and its characterization. In many cases, laboratories and clinics where the therapy will be performed are in different and far-flung facilities, and the cells must therefore be shipped by a courier. The authors evaluated the effects of different storage conditions, in terms of temperature, time of storage and storage solutions on cell viability, cell growth, differentiation potential and molecular characteristics. The aim was to state the most appropriate storage conditions for transporting adipose tissue and Wharton’s jelly-derived stromal cells, ensuring the maintenance of the stemness features for therapeutic application in horses.

**Abstract:**

To use Mesenchymal Stromal Cells (MSCs) in equine patients, isolation and expansion are performed in a laboratory. Cells are then sent back to the veterinary clinic. The main goal of storage conditions during cell transport is to preserve their biological properties and viability. The aim of this study was to evaluate the effects of storage solutions, temperature and time on the characteristics of equine adipose tissue and Wharton’s jelly-derived MSCs. We compared two different storage solutions (plasma and 0.9% NaCl), two different temperatures (4 °C and room temperature) and three time frames (6, 24, 48 h). Cell viability, colony-forming units, trilineage differentiation, the expression of CD45 and CD90 antigens and adhesion potentials were evaluated. Despite the molecular characterization and differentiation potential were not influenced by storage conditions, viability, colony-forming units and adhesion potential are influenced in different way, depending on MSCs sources. Overall, this study found that, despite equine adipose tissue MSCs being usable after 24 h of storage, cells derived from Wharton’s jelly need to be used within 6 h. Moreover, while for adipose cells the best conservation solutions seems to be plasma, the cell viability of Wharton’s jelly MSCs declined in both saline and plasma solution, confirming their reduced resistance to conservation.

## 1. Introduction

During the last twenty years, Mesenchymal Stromal Cells (MSCs) have received, both in human and veterinary medicine, considerable attention because of their potential use for promoting tissue regeneration, not only because of their differentiation potential, but likely also because of their trophic, anti-inflammatory and immunomodulatory abilities [1,2].

In the last twenty years, in equine medicine, bone marrow has been the most used source of autologous MSCs. Alternatively, adipose tissue-derived MSCs have been used [3]. However, for recovering these tissues an invasive procedure is required and a large variability in the cell yield related to the donor has been demonstrated, for both eBM (equine bone marrow) and eAT (equine adipose tissue) [4]. As demonstrated by different authors [5,6,7,8,9], fetal adnexa represent an important source of MSCs for equine regenerative medicine. These tissues can be easily procured without invasive procedures, both for mare and foal, and fetal adnexa-derived MSCs preserve some characteristics typical of primitive native layers and have been defined as an intermediate between embryonic and adult MSCs [10]. In equine, among fetal adnexal tissues, the major source of MSCs is eAM (equine amniotic membrane) [5] and Wharton’s jelly (eWJ) [11]. Recently, Iacono et al. [12], comparing eAMMSCs and eWJMSCs, found that cells isolated from different matrices have different morphological and molecular features and different differentiation potentials. Particularly, data recovered by the Authors show that eWJ could be considered as a more viable and convenient MSCs source for autologous or allogeneic regenerative therapies.

Due to the results obtained after the in vivo use of MSCs in equines, the interest of horse owners and veterinarians is progressively increasing. For in vivo use, MSCs are cultured, expanded and prepared in a specialized laboratory, then they are transferred to the clinic. However, most of the time, the laboratory and the clinic are located far from each other, and some authors hypothesize that one of the reasons why only 24% of the injected cells are found in the lesion site after 24 h is the reduced cell viability resulting from the transposing of cells over long distances [13,14].

In this context, different authors have attempted to find the best conditions (media, temperature, hours of transport, etc.) for shipping equine MSCs from laboratory to clinic. In 2012, Bronzini et al. [15] analyzed the influence of media, temperature and hours of storage on equine peripheral blood MSCs, finding that neither different media nor temperature were able to maintain cell viability during shipping period; in fact, cell mortality was around 30–40% in every experimental group. The same was demonstrated successively by Mercati et al. [16], by Espina et al. [17] and Garvican et al. [14] on equine adipose tissue MSCs, bone marrow and ePBMSCs (equine peripheral blood MSCs), respectively. Stored eATMSCs and eBMMSCs have retained differentiation and clonogenic potential [16,17], while the ability of ePBMSCs to maintain their molecular features is controversial [15]. Indeed, while the level of expression of CD44 and CD105 were constant in fresh and stored cells, CD90 expression decreased after 9 and 12 h of storage, suggesting, as reported by the authors, an inability of these cells to retain the properties of MSCs after this period of storage [15].

Despite the interest in using cells derived from fetal adnexa, eWJMSCs in particular, and the results obtained after their in vivo application [8], to our knowledge no study has reported on the storage and shipping effects on eWJMSCs viability, in vitro growth and molecular features and their differentiation potential. In the present study the effects of storage temperature, time and solution on eWJMSCs characteristics are determined. In order to define which type of cells are the most resistant in case long-distance transport is needed, in the present study the effects of shipping conditions on adult and Wharton’s jelly-derived MSCs have been determined.

## 2. Materials and Methods

Chemicals were obtained from Sigma-Aldrich (Merck); type I collagenase, DMEM (Dulbecco’s modified Eagle’s medium) low glucose medium with Glutamine and FBS (fetal bovine serum) are branded GIBCO (ThermoFisher Scientific, Waltham, MA, USA). Plastics were from Falcon^TM^ unless otherwise stated.

### 2.1. Samples

During the colic surgery of horses spontaneously referred by the owners to the Department of Veterinary Medical Sciences (DIMEVET), University of Bologna, intra-abdominal AT was collected (n = 3). For the use of removed tissue for research purposes owners gave a written consent. Experimental procedures were approved by the Ethics Committee on Animal Use of the University of Bologna (Prot. 55948-X/10).

Wharton’s jelly was isolated from the umbilical cord (UC; n = 3) recovered after foal physiological birth, born from Standardbred mares, and housed at the DIMEVET. For Wharton’s jelly sampling, experimental procedures were approved by the Ethics Committee on Animal Use, University of Bologna (Prot. 55948-X/10), and a written consent was given by the owners to allow tissue recovery for research purposes.

### 2.2. Cell Isolation and Culture

AT and UC samples, stored in DPBS (Dulbecco’s polyphosphate buffer solution) plus antibiotics (100 IU/mL penicillin, 100 g/mL streptomycin), were kept at 4 °C until processing. Under a laminar flow hood, the richest portion of WJ was immediately isolated from the cord tissue.

For both tissues, MSCs were isolated as previously described by Iacono et al. [12]. Briefly, AT and WJ were washed with repeated dives in DPBS, weighed, and cut into 0.5 cm pieces by sterile scissors. Samples were transferred into a 50 mL polypropylene tube and digested by a 0.1% collagenase type I solution in DPBS (1 mL solution/1 g tissue). The suspension was kept in a 37 °C water bath for at least 30 min and vigorously mixed every 10 min. For inactivating the collagenase, the suspension was diluted 1:1 with DPBS plus 10% FBS. The solution was filtered through a stainless steel strainer for discarding undigested tissue and centrifuged at 470× *g* at 25 °C for 10 min. The pellet was re-suspended in DMEM Low Glucose + 10% FBS + 100 U/mL penicillin + 100 g/mL streptomycin. Cells were plated into 25 cm^2^ culture flasks and incubated in 5% CO_2_ at 38.5 °C, in humidified atmosphere (Passage 0). After 48 hrs, the culture medium was completed replaced and non-adherent cells were removed. Culture medium was changed every three days until cell growth reached 80 to 90% confluence. At 80–90% of confluence, cells were dissociated using a 0.25% trypsin EDTA (Ethylenediaminetetraacetic acid) solution and counted and cryopreserved as described by Merlo et al. [18]. Briefly, cells in 0.5 mL of FBS were put in a 1.5 mL cryogenic tube (Sarstetd Inc., Nümbrecht, Germany) at 4 °C. After 10 min, cell suspension was diluted 1:1 with FBS + 16% DMSO (Dimethyl sulfoxide; final concentration 8% DMSO) and maintained for a further 10 min at 4 °C. Then the cryogenic tube was set to −80 °C for 24 h in a “Mr. Frosty” (Nalgene) and finally stored in liquid nitrogen. AT and WJMSCs were thawed at 37 °C in 20 mL DMEM + 10% FBS, then centrifuged at 470 g at 25 °C for 10 min. The pellet was re-suspended in 1 mL of culture medium and cell concentration and viability were evaluated by staining cells with 4% eosin solution and using a Neubawer improved chamber. Cells were plated in a 25 cm^2^ flask (5000 cells/cm^2^) as “Passage 1” (P1).

### 2.3. Study Design

When the confluence of 80–90% at P3 was reached, cells from all three AT and WJ samples were detached from the flask and viability and concentrations were determined as described above. For studying the effects of transport conditions, 6 × 10^6^/mL live AT and WJ cells were stored in equine plasma (P) or 0.9% NaCl solution (S), tested for in vivo use, for 6 (T6), 24 (T24) and 48 (T48) hours at refrigeration (4 °C) and room temperature (RT = 20 °C).

At any time, viability was determined by eosin stain and cells were sown to determining colony forming unit ability, adhesion, and tri-lineage in vitro differentiation potential. Furthermore, CDs expression by PCR was also determined. Un-stored cells, namely T0, were considered as control. For all samples (3 AT and 3 WJ), each test was carried out in three replicates.

### 2.4. CFU (Colony Forming Unit) Assay

For determining the ability of cells to form colonies, 1 × 10^2^ cells at T0, T6, T24 and T48 for both storage conditions were cultured for 8 days in a 30 mm petri dish. Colonies were fixed in 4% paraformaldehyde at RT for 1 h and stained with Giemsa 0.1% stain (15 min). Using an inverted light microscope (Eclipse TE 2000u, Nikon, Tokyo, Japan), the operator counted colonies formed by at least 16–20 nucleate cells.

### 2.5. Spheroid Formation Assays

To determine whether stored cells preserved their adhesion capability, spheroid formation was performed. Differently from the cell-substratum adhesion, performed on monolayer cultures adherent to rigid substrates, this test gives information about the direct cell–cell adhesion architecture found in normal tissues.

Cells were cultured in a multiwell Corning 96-well Black/Clear Round Bottom Ultra- Low Attachment Spheroid Microplate (5000 cells/25 μL drop). Images were acquired after 24 and 48 hrs of culture by a CCD camera (DS-Fi2, Nikon, Tokyo, Japan) mounted on an inverted light microscope (Eclipse TE 2000u, Nikon, Tokyo, Japan).

### 2.6. Multi Lineage In Vitro Differentiation

The osteogenic, adipogenic, and chondrogenic in vitro differentiation potential of control and stored AT- and WJ-MSCs were determined. As reported in Table 1, 5000 cells/cm^2^ were cultured for two weeks under specific induction media. The same number of cells was cultured in culture medium, ss negative control.

For differentiation evaluation, cells were fixed with 4% paraformaldehyde at RT for 1 h; Oil Red O, Alcian Blue, and Alizarin Red were used for staining adipogenic vacuoles, deposits of glycosaminoglycans, and calcium, respectively. An inverted light microscope (Eclipse TE 2000u, Nikon, Tokyo, Japan) was used to observe stained cells.

### 2.7. RT-PCR

For molecular characterization, RNA was extracted from snap-frozen cells and using Nucleo Spin^®^ RNA kit (Macherey-Nagel) following the manufacturer’s instructions. cDNAs were synthesized by RevertAid RT Kit and used directly in PCR reactions, following the instructions of Maxima Hot Start PCR Master Mix.

The expression of genes coding for MSC marker, CD90, and hematopoietic markers CD45 was determined. To ensure the proper expression of samples, GAPDH was used as a housekeeping gene. Primers are listed in Table 2. PCR products were visualized with ethidium bromide on a 2% (*w*/*v*) agarose gel.

### 2.8. Statistical Analysis

Data were analyzed for normal distribution, using a Shapiro–Wilk test. One-way ANOVA, followed by Student–Newman–Keuls’ test when F values indicated significance, was used for analyzing the mean number of colonies and cell viability. Cell viability and CFU are expressed as mean ± SD. Statistical analyses were performed using IBM SPSS Statistics 25 (IBM Corporation). Significance was assessed for *p* < 0.05.

## 3. Results

### 3.1. Vitality and CFU Assay

The mean ± SD of eATMSCs and eWJMSCs at T0 were 90 ± 0% and 95 ± 5%. No statistically significant differences (*p* > 0.05) were observed between vitality at T0 and T6 for eATMSCs stored in saline solution at 4 °C (67.3 ± 24.1%) and in plasma at 4 °C (75.7 ± 14.4%) and RT (71.6 ± 11.5%). Only cells stored for 6 hrs in saline solution at RT showed a significantly lower vitality (19.0 ± 22.6%; *p* < 0.05). At T24, a statistically significant difference was found only in the viability of cells stored in plasma at 4 °C (45.8 ± 12.6%) and cells stored in saline solution at RT (17.8 ± 19.9%; *p* < 0.05). The same was also observed at T48 (P4vsS20: 21.4 ± 6.3% vs. 2.2 ± 3.3%; *p* < 0.05). At 48 the same difference was found also between cells stored in plasma at RT and saline solution at RT (25.1 ± 19.9 vs. 2.2 ± 3.3%; *p* < 0.05). Data are shown in Figure 1.

Data regarding eWJMSCs viability after storage are shown in Figure 2. Different from eATMSCs, all storage groups showed a statistically different viability from T0 (*p* < 0.05), but no differences were observed between groups stored for 6–24–48 hrs in plasma and saline solution at 4 °C and RT (*p* > 0.05).

Stored eATMSCs and eWJMSCs also preserved the ability to form CFU when cultured in vitro; in both cell lines this statistically decreased as storage hrs increased, except for eATMSCs stored for 6 hrs in saline solution at 4 °C. Data are shown in Figure 3 and Figure 4.

### 3.2. Spheroid Formation Assays

eATMSCs and eWJMSCs at T0 were able to form spheroids when cultured for 24 hrs in a multi-well Corning 96-well Black/Clear Round Bottom Ultra-Low Attachment Spheroid Microplate (Figure 5A,B). Stored eATMSCs were able to form spheroids only when they have been preserved for 6 hrs in saline solution, both at 4 °C and RT (Figure 5C). The same cells, stored for 24 and 48 hrs in saline solution RT, after 24 hrs of hanging drop in vitro culture, formed small, separated spheroids (Figure 5E,G). On the other hand, cells from all AT samples, which were stored for the 6–24–48 hrs in plasma at 4 °C and RT, did not form spheroids; after 24 hrs of culture, in a multi-well Corning 96-well Black/Clear Round Bottom Ultra- Low Attachment Spheroid Microplate, they adhered to the plate, returning to the spindle-shape (Figure 5I).

The same was observed also for eWJMSCs stored in plasma for 6–24–48 hrs at 4 °C and RT (Figure 5L). Unlike eATMSCs, eWJMSCs after storage in saline solution for 6–24–48 hrs at 4 °C and RT were not able to form compacted spheroids, as shown in Figure 5D,F,H. In particular, while after 6 hrs of storage, cells were still able to form small and non-compact spheroids, at T24 and T48, cells were round and in suspension.

### 3.3. Multi Lineage In Vitro Differentiation and RT-PCR

As requested by ICST [23], in the present study, in order to evaluate the ability of stored cells to preserve MSCs’ characteristics, we cultured cells in osteogenic, adipogenic, and chondrogenic induction medium for at least 15 days.

Despite the decline in vitality and in the spheroid formation ability as the number of storage hours increased, both eATMSCs and eWJMSCs maintained the ability to differentiate in vitro toward osteogenic, adipogenic, and chondrogenic lineages at any timepoint. In fact, similar to cells at T0, after culturing in induction medium and staining with Alcian Blue, Oil red O, and Alizarin red, stored eATMSCs and eWJMSCs were able to differentiate and accumulate glycosaminoglycan, calcium, and lipid droplet deposition, as demonstrated in Figure 6 and Figure 7, respectively.

At P3 of in vitro culture, fresh cell populations (T0) expressed MSC-associated markers, CD90, but were negative for the hematopoietic marker, CD45. After storage, at any point in time and in both storage solution and temperature, the molecular characterization was preserved by eWJMSCs and eATMSCs.

## 4. Discussion

Equine Mesenchymal Stromal Cells are increasingly used for clinical application. For their isolation and expansion, a laboratory is mandatory. From the laboratory, cells are then sent back to attending clinicians. Preserving MSCs characteristics en route from the laboratory to the clinic is fundamental for the success of the therapy. Due to the importance of this topic for equine regenerative medicine, in the last 10 years, different storage solution, temperatures, and hours have been tested [14,15,16,17]. Despite the increasingly recognized importance of MSCs derived from Wharton’s jelly and fetal adnexa in the context of equine regenerative therapy, no papers are present in the literature on the best choice for their storage. Moreover, before now, in addition to plasma, authors tested only solution authorized for in vitro use, such as PBS or FBS.

In this context, in the present paper, the effects of saline solution authorized for intravenous administration and equine plasma at 4 °C and RT for 6–24–48 hrs were tested on eATMSCs and on eWJMSCs for the first time.

Unlike the data reported by Garvican et al. [14], in our storage conditions, cell vitality appeared reduced after 6 hrs of storage in both eAT and eWJMSCs. Particularly, cells derived from Wharton’s jelly seem to be more sensitive to storage, especially at refrigeration temperatures; indeed, after 6 hrs of storage, eWJMSCs viability is higher for cells maintained in saline solution and plasma membranes at RT. Bronzini et al. [15] observed the same using ePBMSCs, while eATMSCs seem to withstand refrigeration temperature storage better, as already stated by Mercati et al. [16]. However, after 6 hrs of storage cell vitality was reduced in all study groups, in line with data already reported by different authors [14,15,16,17].

Observing the data of cell vitality, plasma seems to be the best solution for shipping cells from the laboratory to the clinic. However, the number of CFUs recorded both for eATMSCs and eWJMSCs stored in plasma are lower than that recorded for cells stored in saline solution. Moreover, cells stored in plasma, both at 4 °C and RT, are unable to form spheroids. As reported by Iacono et al. [12], the ability to form spheroids is related to the in vitro differentiation potential in the cartilaginous sense. In the present study, cells stored in plasma lost the ability to form spheroids but retained their potential to differentiate in vitro toward three lineages, as requested by ISCT [23]. These findings could be related to in vitro reaction between plasma and culture medium; in fact, in the culture plates of cells stored in plasma, after 24 hrs of in vitro culture, we observed the formation of a fibrin clot, in which the cells are most likely trapped without being able to form colonies and spheroids. The formation of clots could be avoided by adding heparin to plasma for shipping. However, the impact of heparin and its concentration could be studied in eATMSCs and WJMSCs. Indeed, in human MSCs doses of heparin between 100 and 1000 µg/mL of culture medium inhibit cell growth and changes in gene expression [24].

As a quality control tests, we also used in vitro differentiation and molecular characterization. As previously reported by Mercati et al. [16] for eATMSCs, the present study confirms that for both eATMSCs and eWJMSCs maintained their differentiation potential in vitro.

In the storage conditions used in this study molecular characterizations have also been preserved. Indeed, both eATMSCs and eWJMSCs expressed mesenchymal markers, CD90, and were negative for hematopoietic markers, CD45, as well as control group T0, unlike in ePBMSCs observed by Bronzini et al. [15], indicating that storage conditions can probably modify the characteristics of MSCs in different ways depending on their origin.

As previously reported, data reported in the present study confirm that MSC therapy could be administered as soon as possible after cell preparation. Different from results reported by Mercati et al. [16], in the case of therapy with eWJMSCs, this has to be administered within 6 hrs, while for eATMSCs, despite a decrease in vitality at 24 h of storage, it is possible to use them 24 hrs after their preparation. Despite the molecular characterization and differentiation potential were not influenced by storage conditions, both in eWJMSCs and eATMSCs, viability, CFU, and adhesion potential are influenced in different way, depending on MSCs sources. Overall, this study stated that, despite eATMSCs being able to be used after 24 hrs of storage, eWJMSCs need to be used within 6 hrs. Moreover, while for eATMSCs the best conservation solutions seems to be plasma, cell viability of eWJMSCs declined in both saline and plasma solution, confirming their reduced resistance to conservation.

Data recovered in vitro in the present study need to be compared with results obtained in vivo using cells shipped under tested conditions and with data obtained using frozen cells implanted directly, immediately after thawing.

## 5. Conclusions

In the present study we demonstrated that different types of MSCs react differently to the storage conditions frequently used for shipping them from the laboratory to the clinic. These conditions influence viability and, depending on the cell type, they can also influence different MSCs characteristics. Particularly, according to the data recovered in the present study, eWJMSCs need to be used quickly to maintain their vitality characteristics.

In conclusion, as different MSCs doses are being shipped every day, further studies are necessary to find the best shipping conditions for each cell type.

## Figures and Tables

**Figure 1 animals-12-01967-f001:**
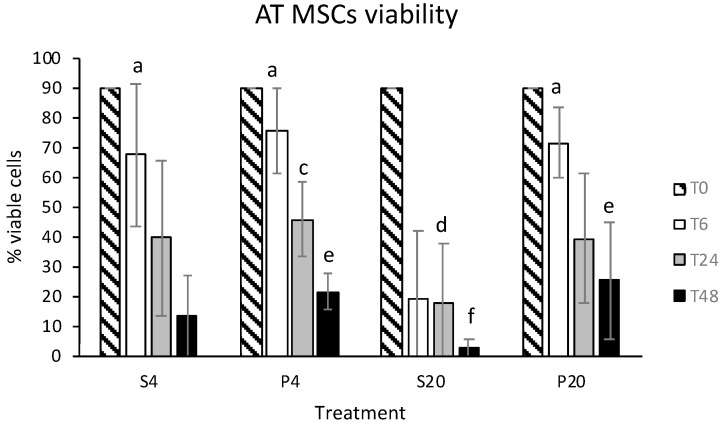
eATMSCs viability after storage for 6 (T6), 24 (T24) and 48 (T48) hrs in saline solution (S) and plasma (P) at 4 °C (4) and room temperature (20): a vs. b; c vs. d; e vs. f: *p* < 0.05.

**Figure 2 animals-12-01967-f002:**
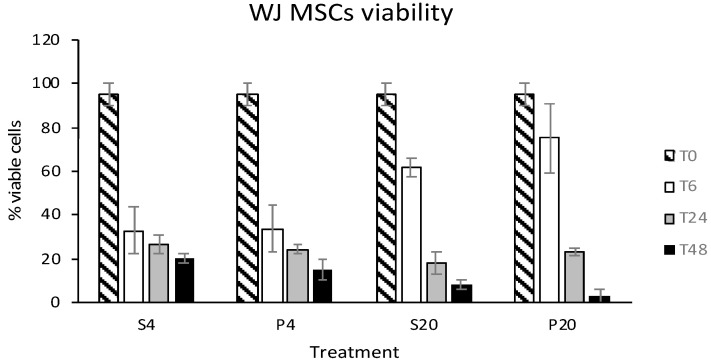
eWJMSCs viability after storage for 6 (T6), 24 (T24) and 48 (T48) hrs in saline solution (S) and plasma (P) at 4 °C (4) and room temperature (20). No statistical differences have been found among groups.

**Figure 3 animals-12-01967-f003:**
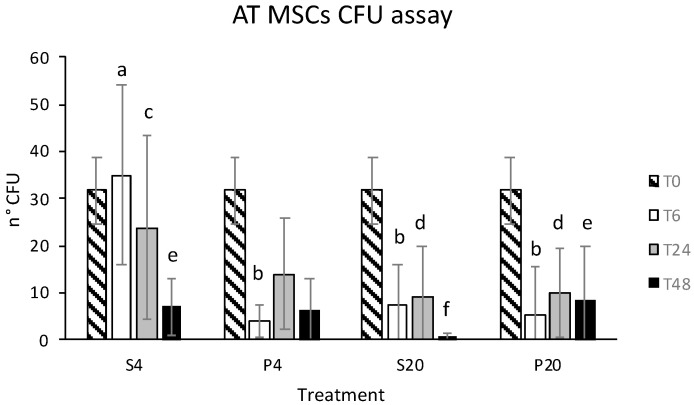
Number of CFU formed by eATMSCs after storage for 6 (T6), 24 (T24) and 48 (T48) hrs in saline solution (S) and plasma (P) at 4 °C (4) and room temperature (20): a vs. b; c vs. d; e vs. f: *p* < 0.05.

**Figure 4 animals-12-01967-f004:**
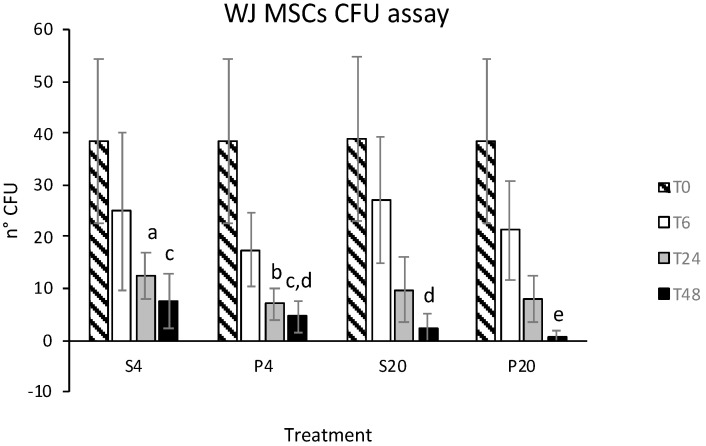
Number of CFU formed by eWJMSCs after storage for 6 (T6), 24 (T24), and 48 (T48) hrs in saline solution (S) and plasma (P) at 4 °C (4) and room temperature (20): a vs. b; c vs. d vs. e: *p* < 0.05.

**Figure 5 animals-12-01967-f005:**
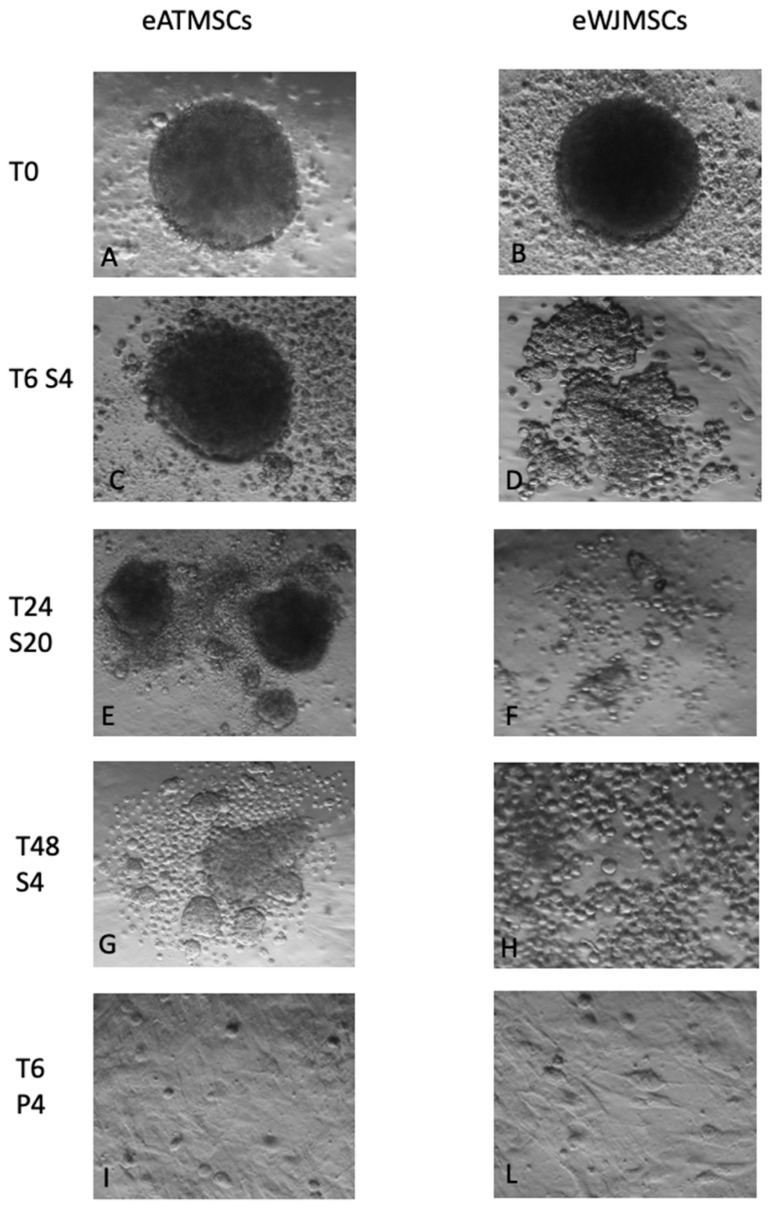
Spheroid formation assay performed with eATMSCs and eWJMSCs. (**A**,**B**) spheroids at T0. As showed in the pictures eATMSCS stored for 6 hrs in saline solution were able to form spheroids (**C**), but after 24 and 48 hrs of storage they formed smaller and fragmented spheroids (**E**,**G**). On the contrary in saline solution stored eWJMSCs were not able to form spheroids (**D**,**F**,**H**). Both cell lines stored in plasma lost their ability to form spheroids and grew adherent to plastic with a fibroblast like form (**I**,**L**). Magnification 10×.

**Figure 6 animals-12-01967-f006:**
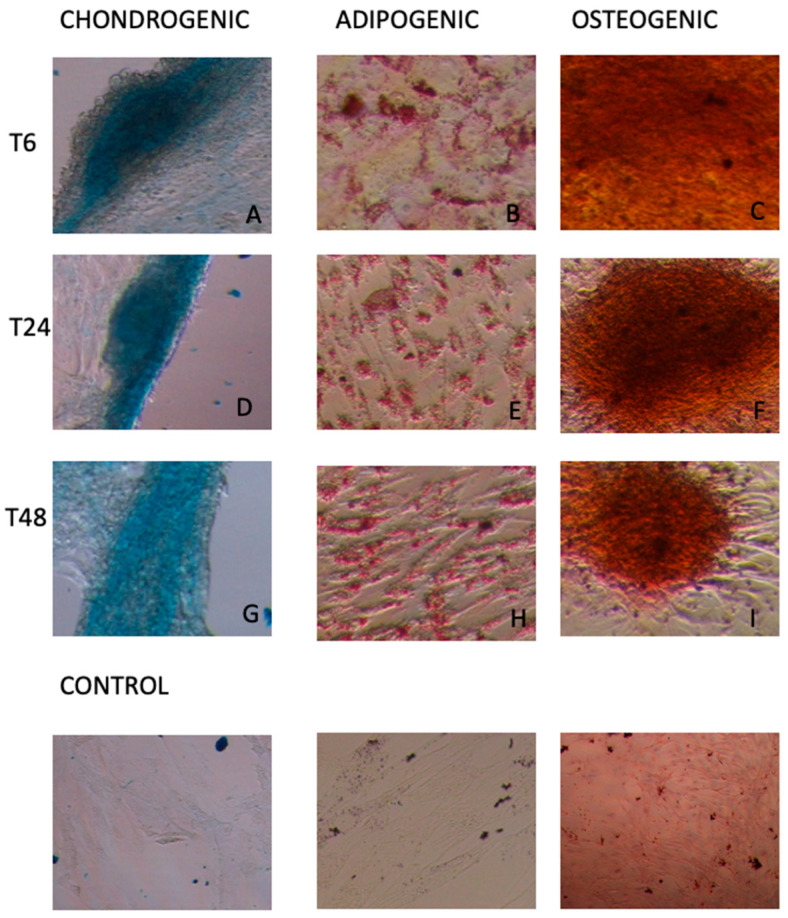
Stored eATMSCs, cultured in induction medium for two weeks and stained for chondrogenic, adipogenic and osteogenic differentiation. T6: (**A**) cells stored in plasma at 4 °C; (**B**) cells stored in plasma at RT; (**C**) cells stored in saline solution at RT. T24: (**D**) cells stored in saline solution at 4 °C; (**E**) cells stored in plasma at 4 °C; (**F**) cells stored in plasma at RT. T48: (**G**) cells stored in saline solution at RT; (**H**) cells stored in saline solution at 4 °C; (**I**) cells stored in plasma at RT. Control groups are reported. Magnification 10×.

**Figure 7 animals-12-01967-f007:**
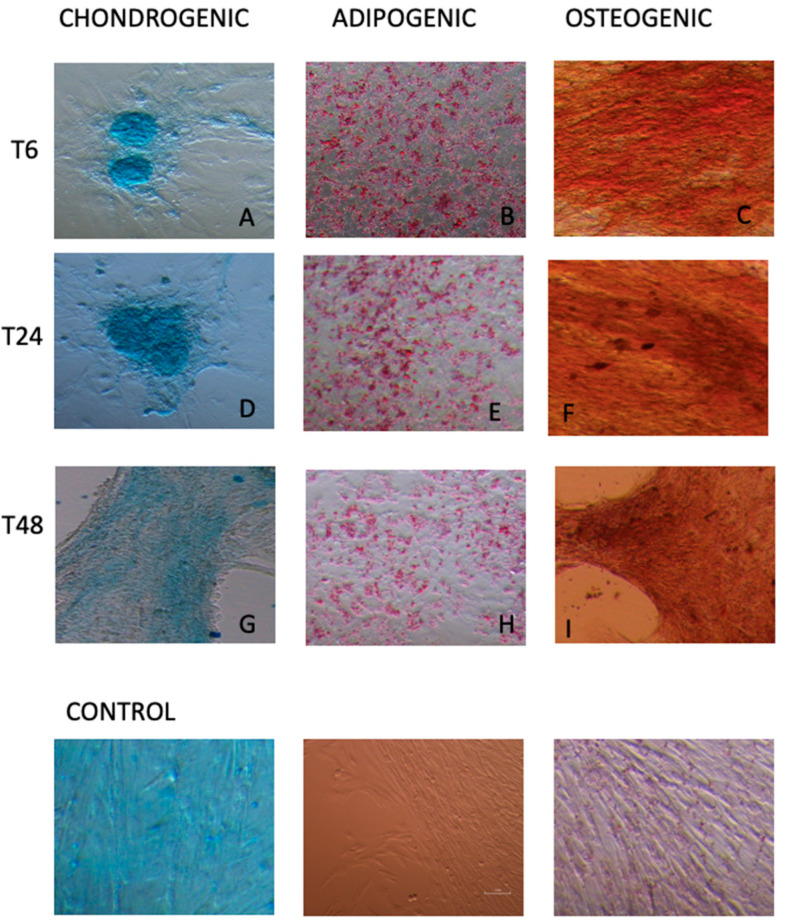
Stored eWJMSCs, cultured in induction medium for two weeks and stained for chondrogenic, adipogenic and osteogenic differentiation. T6: (**A**) cells stored in plasma at 4 °C; (**B**) cells stored in plasma at RT; (**C**) cells stored in saline solution at RT. T24: (**D**) cells stored in saline solution at 4 °C; (**E**) cells stored in plasma at 4 °C; (**F**) cells stored in plasma at RT. T48: (**G**) cells stored in saline solution at RT; (**H**) cells stored in saline solution at 4 °C; (**I**) cells stored in plasma at RT. Control groups are reported. Magnification 10×.

**Table 1 animals-12-01967-t001:** Composition of induction media [19,20].

Adipogenic	Chondrogenic	Osteogenic
DMEM	DMEM	DMEM
10% FBS	1% FBS	10% Rabbit Serum
0.5 mM IBMX (removed after 3 days)	6.25 μg/mL insulin	50 μM AA2P
1 μM DXM (removed after 6 days)	50 nM AA2P	0.1 μM DXM
10 μg/mL insulin	0.1 μM DXM	10 mM BGP
0.1 mM indomethacin	10 ng/mL hTGF-β1	

IBMX: isobutylmethylxanthine, DXM: dexamethasone, hTGF: human transforming growth factor, AA2P: ascorbic acid 2-phosphate, BGP: beta-glycerophosphate.

**Table 2 animals-12-01967-t002:** Primers sequences for PCR analysis.

Primers	References	Sequences (5′→3′)	bp
**MSC marker**			
CD90	[21]	FW: TGCGAACTCCGCCTCTCT	93
		RW: GCTTATGCCCTCGCACTTG	
**Ematopoietic markers**			
CD45	[21]	FW: TGATTCCCAGAAATGACATGTA	101
		RW: ACATTTTGGGCTTGTCCTTAAC	
**Housekeeping**			
GAPDH	[22]	FW: GTCCATGCCATCACTGCCAC	262
		RW: CCTGCTTCACCACCTTCTTG	

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
