# Peer review of "Shipping Temperature, Time and Media Effects on Equine Wharton’s Jelly and Adipose Tissue Derived Mesenchymal Stromal Cells Characteristics"

_animals, 2022, doi:10.3390/ani12151967_

Round 1

Reviewer 1 Report

The paper presents for the first time, the effects of saline solution, authorized for intravenous administration, and equine plasma at 4°C and RT, for 6-24-48 hrs, on eATMSCs and eWJMSCs.

Generaly the topic is valid and practically interesting for the vet docs. 

Please answer or comment on questions listed below:

1.      For molecular characterization control the Expression of genes coding for MSC marker, CD90, and hematopoietic markers CD45, was determined. Why only these two? What about CD44, CD105?

2.      2.Repetition- line 302 more sensitive more sensitive to storage, especially at refrigeration …

3.   3.   “Despite by observing data of vitality plasma seems to be the best solution for shipping cells from 308 the laboratory to the clinic, in the present study the number of CFU, both for eATMSCs and eWJMSCs 309 are lower than that recorded for cells stored in saline solution. Moreover, cells stored in plasma, both 310 at 4°C and RT, are unable to form spheroids. In our opinion, these findings could be related to in vitro 311 reaction between plasma and culture medium. In fact, in the culture plates of cells stored in plasma, 312 after 24 hrs of in vitro culture, we observed the formation of a fibrin clot in which the cells are most 313 likely trapped without being able to form colonies and spheroids.”

 This is a mistake- if CFU is lower  so cells are losing their proliferation ability.

If there is a lack of spheroids formation so cells lost their multipotential characteristics.

Plasma is worse than saline solution. Formation of fibrin clot is forbidden. In case of plasma you should have added heparin for shipping, that would prevent formation of clot.

 4.      Figure 1: Spheroid formation assay.- plasma what time and  temperature? Please describe better the presented fig. (A-L)

5.      Figure 2: Stored eATMSCs, cultured in vitro in induction medium for two weeks and 277 stained for chondrogenic, adipogenic and osteogenic differentiation. Control groups are re-278 ported. Magnification 10x. 279 – I see temperatures but it is not known what solution

6.      The same refers to fig 3.

 7.      Disagree with the statement- using frozen cells, 332 resuspension in a solution for in vivo use could be mandatory but, if it is performed without a laminar 333 flow hood, it could lead to sample contamination by environmental pollutants (virus, bacteria, etc.), 334 even if these cells are produced and expanded in a GMP laboratory, as well as the commercial prod-335 uct [25,26].

Sample contamination by environmental pollutants (virus, bacteria, etc.),is more possible when cells are transported without freezing. I would say that keeping cells without standard in vitro conditions (incubator etc.) over 12hs is more risky than freezing and thawing after reaching the desired destination.

8.      The following fragment from the abstract should be repeated in the conclusions

 „Despite the molecular characterization and 25 differentiation potential were not influenced by storage conditions, both in eWJMSCs and eATMSCs, viability, 26 CFU, and adhesion potential are influenced in different way, depending on MSCs sources. Overall, this study 27 stated that, despite eATMSCs could be used after 24 hrs of storage, eWJMSCs need to be use within 6 hrs. More-28 over, while for eATMSCs the best conservation solutions seems to be plasma, cell viability of eWJMSCs declined 29 in both saline and plasma solution, confirming their their reduced resistance to conservation. 30”

Grapgics 1 to 4  are not sufficiently described. Abreviations should be explained, statistical differences  should be included. 

Author Response

The authors thank the reviewer for comments and suggestions.

In red the answers of the authors

Reviewer 1:

The paper presents for the first time, the effects of saline solution, authorized for intravenous administration, and equine plasma at 4°C and RT, for 6-24-48 hrs, on eATMSCs and eWJMSCs.

Generally the topic is valid and practically interesting for the vet docs.

Please answer or comment on questions listed below:

1.For molecular characterization control the Expression of genes coding for MSC marker, CD90, and hematopoietic markers CD45, was determined. Why only these two? What about CD44, CD105? The cells used in the present study had already been previously characterized. In this study we therefore decided to verify only if the conservation determined a modification of the expression of a typical CD of stemness and a hematopoietic CD.

.

  1. Repetition- line 302 more sensitive more sensitive to storage, especially at refrigeration: the authors corrected as suggested by the reviewer

3.“Despite by observing data of vitality plasma seems to be the best solution for shipping cells from 308 the laboratory to the clinic, in the present study the number of CFU, both for eATMSCs and eWJMSCs 309 are lower than that recorded for cells stored in saline solution. Moreover, cells stored in plasma, both 310 at 4°C and RT, are unable to form spheroids. In our opinion, these findings could be related to in vitro 311 reaction between plasma and culture medium. In fact, in the culture plates of cells stored in plasma, 312 after 24 hrs of in vitro culture, we observed the formation of a fibrin clot in which the cells are most 313 likely trapped without being able to form colonies and spheroids.”

 This is a mistake- if CFU is lower  so cells are losing their proliferation ability. In our opinion cells didn’t lose their proliferation ability, but the problem was the fibrin clot in which the cells have become trapped. In fact, those who managed to get out were able to form colonies and then replicate.

If there is a lack of spheroids formation so cells lost their multipotential characteristics. See highlighted text

Plasma is worse than saline solution. Formation of fibrin clot is forbidden. In case of plasma you should have added heparin for shipping, that would prevent formation of clot. See highlighted text

  1. Figure 1: Spheroid formation assay.- plasma what time and  temperature? Please describe better the presented fig. (A-L). See highlighted text

  1. Figure 2: Stored eATMSCs, cultured in vitro in induction medium for two weeks and 277 stained for chondrogenic, adipogenic and osteogenic differentiation. Control groups are re-278 ported. Magnification 10x. 279 – I see temperatures but it is not known what solution. See highlighted text

  1. The same refers to fig 3. See highlighted text

  1. Disagree with the statement- using frozen cells, 332 resuspension in a solution for in vivo use could be mandatory but, if it is performed without a laminar 333 flow hood, it could lead to sample contamination by environmental pollutants (virus, bacteria, etc.), 334 even if these cells are produced and expanded in a GMP laboratory, as well as the commercial prod-335 uct [25,26].

Sample contamination by environmental pollutants (virus, bacteria, etc.),is more possible when cells are transported without freezing. I would say that keeping cells without standard in vitro conditions (incubator etc.) over 12hs is more risky than freezing and thawing after reaching the desired destination. The sentences in line 332-335 have been eliminated

  1. The following fragment from the abstract should be repeated in the conclusions

 „Despite the molecular characterization and 25 differentiation potential were not influenced by storage conditions, both in eWJMSCs and eATMSCs, viability, 26 CFU, and adhesion potential are influenced in different way, depending on MSCs sources. Overall, this study 27 stated that, despite eATMSCs could be used after 24 hrs of storage, eWJMSCs need to be use within 6 hrs. More-28 over, while for eATMSCs the best conservation solutions seems to be plasma, cell viability of eWJMSCs declined 29 in both saline and plasma solution, confirming their their reduced resistance to conservation. 30” See highlighted text

Grapgics 1 to 4  are not sufficiently described. Abreviations should be explained, statistical differences  should be included. See highlighted text

.

Reviewer 2 Report

Authors try to find the best conditions (media, temperature, hours of transport, exc.) for shipping equine MSCs from laboratory to the clinic. The experiments were well designed. The selections of methods are suitable, and the methods are used appropriately.

I have some minor suggestions:

1.The abbreviations should present in the form of full name in abstract and in the main text for the first time they occurrences.

2. Line136, 37C? same problem in line 138.

3. Line 140 what is for P0?

4. Line 142, re-placeing?

5. Part 2.6, What is the method for relative expression calculation?

6. I don’t know the descriptions of Graphic 1-4 is suitable or instead of figures. Moreover, figure legends should be added at least.

7. Line 293, the order of reference is not correct.

8. Line 295, Authors should be authors.

Author Response

The authors thank the reviewer for comments and suggestions.

In red the answers of the authors

Reviewer 2:

Authors try to find the best conditions (media, temperature, hours of transport, exc.) for shipping equine MSCs from laboratory to the clinic. The experiments were well designed. The selections of methods are suitable, and the methods are used appropriately. 

I have some minor suggestions:

1.The abbreviations should present in the form of full name in abstract and in the main text for the first time they occurrences. the authors corrected as suggested by the reviewer

  1. Line136,37C? same problem in line 138. the authors corrected as suggested by the reviewer
  2. Line 140 what is for P0? the authors corrected as suggested by the reviewer
  3. Line 142, re-placeing? the authors corrected as suggested by the reviewer
  4. Part 2.6, What is the method for relative expression calculation? In the present study we didn’t determine the relative expression calculation, but we use the PCR only to verify if the storage time, temperature and solution modified the total expression of CDs.
  5. I don’t know the descriptions of Graphic 1-4 is suitable or instead of figures. Moreover, figure legends should be added at least. the authors corrected as suggested by the reviewer
  6. Line 293, the order of reference is not correct. the authors corrected as suggested by the reviewer
  7. Line 295, Authors should be authors. the authors corrected as suggested by the reviewer
